# Management of childhood and adolescent latent tuberculous infection (LTBI) in Germany, Austria and Switzerland

Ulrich von Both[1,2], Philipp Gerlach[1], Nicole Ritz[3,4], Matthias Bogyi[5], Folke Brinkmann[6], Stephanie Thee[7] *

1 Division of Pediatric Infectious Diseases, Dr von Hauner Children's Hospital, University Hospital, Ludwig-Maximilian University (LMU), Munich, Germany, 2 German Centre for Infection Research, Partner Site Munich, Munich, Germany, 3 Pediatric Infectious Diseases Unit, University Children's Hospital Basel, The University of Basel, Basel, Switzerland, 4 Department of Pediatrics, The Royal Children's Hospital Melbourne, The University of Melbourne, Parkville, Australia, 5 Department of Paediatrics, Wilhelminenspital, Vienna, Austria, 6 Department of Paediatric Pulmonology, Ruhr University Bochum, Bochum, Germany, 7 Department of Pediatric Respiratory Medicine, Immunology and Critical Care Medicine, Charité – Universitätsmedizin, Berlin, Germany

* stephanie.thee@charite.de

**Data Availability Statement:** All relevant data are within the paper and its Supporting Information files.

## Abstract

### Background

Majority of active tuberculosis (TB) cases in children in low-incidence countries are due to rapid progression of infection (latent TB infection (LTBI)) to disease. We aimed to assess common practice for managing paediatric LTBI in Austria, Germany and Switzerland prior to the publication of the first joint national guideline for paediatric TB in 2017.

### Methods

Online-based survey amongst pediatricians, practitioners and staff working in the public health sector between July and November 2017. Data analysis was conducted using IBM SPSS.

### Results

A total of 191 individuals participated in the survey with 173 questionnaires included for final analysis. Twelve percent of respondents were from Austria, 60% from Germany and 28% from Switzerland. Proportion of children with LTBI and migrant background was estimated by the respondents to be >50% by 58%. Tuberculin skin test (TST) and interferon-γ-release-assay (IGRA), particularly Quantiferon-gold-test, were reported to be used in 86% and 88%, respectively. In children > 5 years with a positive TST or IGRA a chest x-ray was commonly reported to be performed (28%). Fifty-three percent reported to take a different diagnostic approach in children ≤ 5 years, mainly combining TST, IGRA and chest x-ray for initial testing (31%). Sixty-eight percent reported to prescribe isoniazid-monotherapy: for 9 (62%), or 6 months (6%), 31% reported to prescribe combination therapy of isoniazid and rifampicin. Dosing of isoniazid and rifampicin below current recommendations was reported by up to

**Funding:** PG received a travel grant from Ludwig Maximilian University Munich, Germany. The funders had no role in study design, data collection and analysis, decision to publish, or preparation of the manuscript.

**Competing interests:** The authors have declared that no competing interests exist.

22% of respondents. Blood-sampling before/during LTBI treatment was reported in >90% of respondents, performing a chest-X-ray at the end of treatment by 51%.

## Conclusion

This survey showed reported heterogeneity in the management of paediatric LTBI. Thus, regular and easily accessible educational activities and national up-to-date guidelines are key to ensure awareness and quality of care for children and adolescents with LTBI in low-incidence countries.

## Introduction

In 2014, the World Health Organization (WHO) released an action framework to eliminate tuberculosis (TB) in low incidence countries [1] as part of the global "Stop TB" strategy. This action framework aims to progress from <100 cases per 1 million inhabitants towards "pre-elimination" of TB (defined as <10 TB cases per million) until 2035 [1]. Because in low-incidence countries most cases of active TB in children are due to rapid progression of latent TB infection (LTBI) to TB disease, prevention strategies must focus on early screening for and treatment of LTBI. This is particularly true for migrant children who recently arrived from high-TB-endemic regions [2, 3]. Following infection with *Mycobacterium tuberculosis (M. tb)*., there is a general life-time risk of 5–10% to develop TB [4, 5]. In children and adolescents however, the risk to develop TB disease following infection is far greater and estimated to be 15% within 5 years up to 33% within one year depending on age and study setting [6]. The detection of LTBI relies on immune-based diagnostics: either the tuberculin skin-test (TST) or an interferon-γ-release assay (IGRA). While IGRA testing has demonstrated a greater specificity than TST, its sensitivity in children younger than 5 years of age is less established [7]. In order to exclude pulmonary TB, a chest x-ray is routinely recommended even in asymptomatic children [8]. Once, TB disease is excluded, highly LTBI treatment regimens exist to prevent progression from infection to TB disease. These regimens include isoniazid (INH) for 6–9 months, rifampicin (RMP) for 4months or a combination therapy with INH and RMP for 3 months [9, 10]. Currently recommended daily doses in TB therapy for INH and RMP are INH 10 mg/kg and RMP 15 mg/kg bodyweight, respectively [11] and were increased from previous guidelines from INH 5mg/kg and RMP 10mg/kg following pharmacokinetic studies in children [12–16]

Austria, Germany and Switzerland represent a population of 100 million people in central Europe with figures of 8.9, 82.6, and 8.5 million, respectively. All three countries are classified as low-TB-incidence countries with an incidence of 5.0–5.8 per 100,000 in 2018/2019 [17–19]. According to national infection protection acts, refugees as well as contacts of persons with infectious tuberculosis are to be investigated for tuberculosis [2, 20]. Only by 2017, first common guidelines on management of paediatric TB including LTBI were published for these three countries [21]. These guidelines emphasize that "intention to test is intention to treat" and any positive immunological test (TST and/or IGRA), should lead to initiation of treatment for LTBI or investigation and treatment for TB disease [21]. In 2016, there was an European-wide shortage of PPD RT23 SSI tuberculin strain (manufactured by the Statens Serum Institut, Copenhagen, Denmark), possibly influencing diagnostic procedures [22]. Because of the increased risk for progression to disease [4, 6], diagnostic procedures and management of LTBI might differ in children younger than 5 years-of age compared to older children.

This study aims to describe clinical practice for LTBI in the paediatric population of Austria, Germany and Switzerland and to add important real-life data to the limited body of evidence on diagnostic procedures, therapy and management in children of different age-groups with LTBI from low-incidence countries.

## Methods

Members of established paediatric pneumology and infectious diseases societies including the Austrian Pediatric Society (ÖGKJ), the Germany-based Society for Pediatric Pneumology (GPP), the German Society for Pediatric Infectious Diseases (DGPI), the Swiss Society for Pediatric Pneumology (SGPP-SSPP) and the Pediatric Infectious Disease Group of Switzerland (PIGS) as well as members of the responsible public health department were invited via email to participate in the survey. In Switzerland, cantonal representatives from the Swiss Lung Association (Lungenliga Schweiz) and university hospitals were contacted for personal interviews and to fill in the survey. The survey consisted of 44 questions divided into 6 subsections and was available in German, English and French (the English version of the document is available, see S1 Appendix).

Data were collected between July 14th until November 15th 2017 via a web-based tool (surveymonkey) creating a standardized dataset for each case or a paper-based questionnaire. It was made sure that no respondent filled in the questionnaire more than once using IP-lock. Information on the affiliated institution, the diagnostic methods including procedures to exclude TB disease, treatment regimens for LTBI and follow-up was collected. This survey sought responses from healthcare professionals via collaborative networks and societies and did not contain any patient identifiable data; ethics approval was therefore not required. Data collected consisted of categorical variables and are presented as frequencies (number and percentage of patients). Data analysis was conducted with IBM SPSS version 25.

## Results

### Study population

A total of 191 individuals took part in the survey of which 18 data sets were exclude because of substantially incomplete data. One-hundred-seventy-three data sets were included in the final analysis of which 104/173 (60.1%) respondents were from Germany, 49/173 (28.3%) from Switzerland and 20/173 (11.6%) from Austria. Number of respondents per federal state ("*Bundesland*" in Germany and Austria and "*Kanton*" in Switzerland) and institutional affiliations of the respondents are depicted in Fig 1 and Table 1, respectively. Seventy-three of 173 (42.2%) respondents reported to evaluate less than 10 children for LTBI/TB each year, 83/173 (48.0%) 10–50 children, and 16/173 (9.3%) more than 50 children per year; one respondent provided no data (0.6%). The respondents estimated the proportion of migrant children to be >80% in 47/173 (27.2%), 50–80% in 54/173 (31.2%), 20–50% in 23/173 (13.3%) and <20% in 10/173 (5.8%), respectively. One respondent (0.6%) reported none of the children being migrants and in 38/173 (22.0%) no information was provided.

### Immunodiagnostic testing

When testing for LTBI, 149/173 (86.1%) reported to perform a TST, 152/173 (87.9%) an IGRA-test. Quantiferon gold (Plus) was reported to be the IGRA-test used by 118/173 (68.2%), T-SPOT.TB by 18/173 (10.4%) and both IGRA-tests equally by 19/173 (11.0%); for 18/173 (10.4%) respondents this data were not provided. Choice of IGRA differed between the three countries: T-SPOT.TB alone was used by 2/20 (10.0%) in Austria, by 9/104 (8.7%) respondents

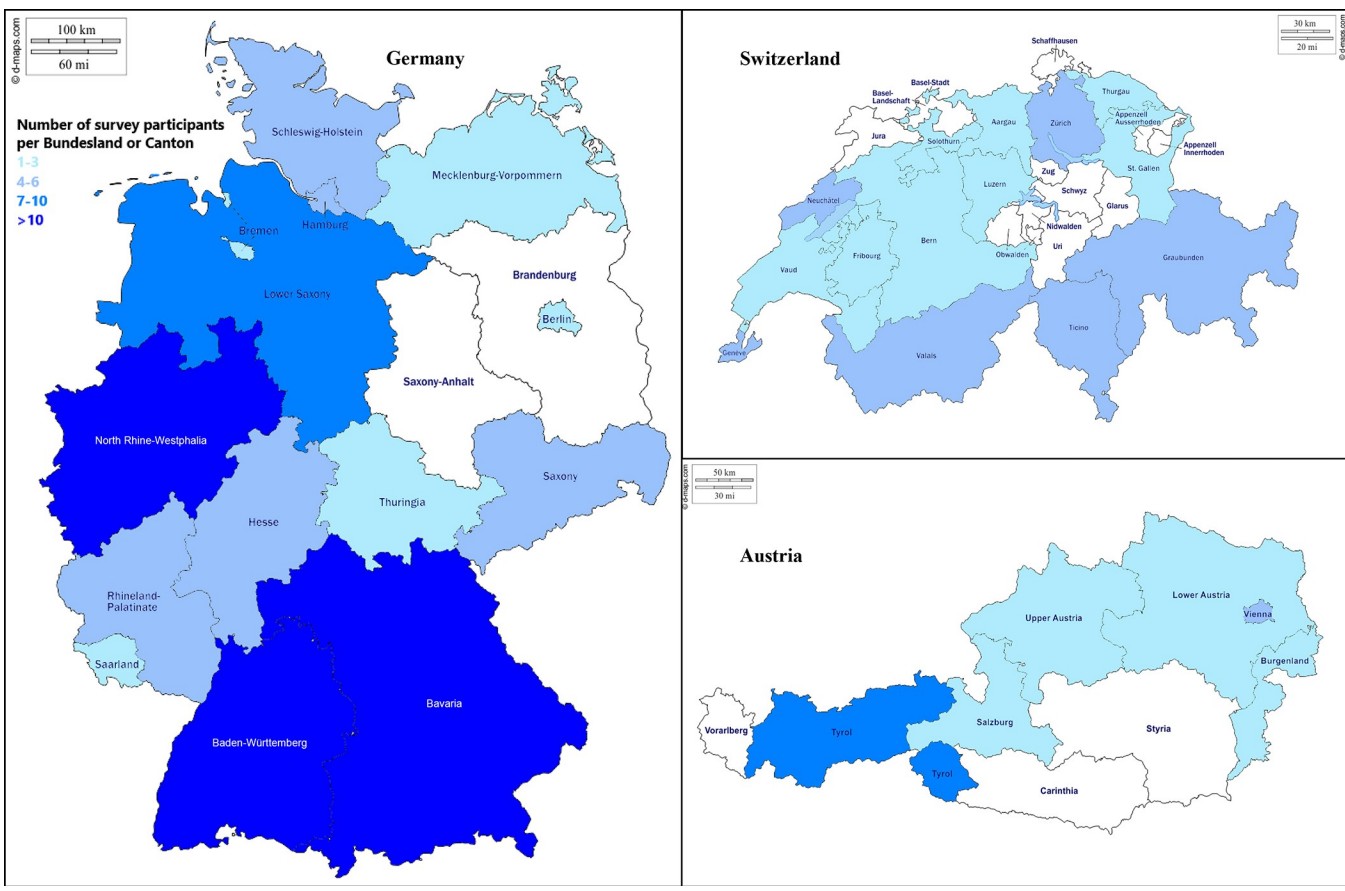

**Fig 1. Number of survey respondents per federal state or canton, respectively.** Reprinted under a CC BY license, with permission from [Daniel Dalet], original copyright [2020].

in Germany, and 7/49 (14.3%) in Switzerland while both IGRA tests were used by 9/104 (8.7%) in Germany, 4/20 (20.0%) in Austria and 6/49 (12.2%) in Switzerland, respectively.

Ninety-one/173 (52.6%) of the respondents reported to have been affected by the 2016 shortage of PPD RT 23 SSI tuberculin. In 18/91 (19.8%) cases an alternative tuberculin was used. Three respondents from Germany (3.3%) stated to rather use TST in younger children, but were forced to switch to IGRA testing due to this PPD RT 23 SSI shortage.

## Radiology

For the diagnosis of LTBI in children > 5 years of age, the most common procedure reported to exclude TB-disease (48/173, 27.7%) was performing a chest-x-ray in case of an either positive IGRA or TST.

**Table 1. Institutional and geographical background of survey respondents.**

| Institution / Country | University hospital | Central hospital / teaching hospital | District general hospital | Private hospital | Private practice | Shared doctor's office or medical care center | Public health service | Total |
|---|---|---|---|---|---|---|---|---|
| **Austria** | 2 | 7 | 3 | 0 | 0 | 0 | 8 | 20 |
| **Germany** | 24 | 27 | 7 | 4 | 11 | 1 | 30 | 104 |
| **Switzerland** | 14 | 14 | 1 | 1 | 5 | 0 | 14 | 49 |
| **Total** | 40 | 48 | 11 | 5 | 16 | 1 | 52 | 173 |

**Table 2. Reported diagnostics for LTBI in children older and younger than 5 years of age.**

| Procedure | > 5 years of age | ≤5 years of age |
|---|---|---|
| **TST only** | 5 (2.9%) | 3 (1.7%) |
| **IGRA-test only** | 7 (4.0%) | 2 (1.2%) |
| **TST + IGRA** | 9 (5.2%) | 6 (3.5%) |
| **TST + chest-x-ray** | 5 (2.9%) | 41 (23.7%) |
| **IGRA + chest-x-ray** | 29 (16.8%) | 5 (2.9%) |
| **TST + IGRA + chest-x-ray** | 37 (21.4%) | 54 (31.2%) |
| **Chest-x-ray only if IGRA or TST positive** | 48 (27.7%) | 22 (11.7%) |
| **Other** | 13 (7.5%) | 18 (10.4%) |
| **Missing/unknown** | 20 (11.6%) | 21 (12.2%) |

Ninety-three/173 (53.8%) respondents reported to choose a different approach for children ≤5 years of age. One hundred/173 (57.8%) respondents reported to include a chest x-ray in the initial diagnostic work-up for LTBI in children ≤5 years of age compared to 71/173 (41.0%) in children > 5 years of age, respectively. Table 2 depicts diagnostic procedures and their frequencies in both age groups.

## Treatment

For LTBI treatment, information on treatment was missing in 29/173 (16.7%) or unknown in 5/173 (2.9%) respondents. In the remaining 139 answers, 94/139 (67.6%) reported to prescribe INH monotherapy: 87/139 (62.6%) used INH monotherapy as their first choice, whereby in one respondent a different approach was chosen for migrant children and in another respondent INH monotherapy was only prescribed for children ≤ 5 years of age (and combination of INH and RMP in children >5years); 4/139 (2.8%) stated to equally use INH or RMP and 3/139 (2.2%) either INH or a combination of INH and RMP. Two/139 (1.4%) used RMP monotherapy (both for four months), 43/139 (30.9%) a combination of INH and RMP. Table 3 describes duration and dose of commonly used anti-TB agents. Of note, LTBI treatment was exclusively prescribed as daily therapy with no intermittent regimens reported. For INH, 9/94 (9.6%) respondents reported doses below the recommended range, and 12/54 (22.2%) for RMP. INH monotherapy was reported to be prescribed by respondents from Germany in 52/76 (68,4%), by respondents from Austria in 8/18 (44,4%) and from Switzerland in 27/45 (60,0%), while for combination therapy of INH and RMP it was 22/76 (28.9%), 6/18 (33.3%) and 15/45 (33.3%), respectively.

## Monitoring and follow-up during treatment

Information on monitoring of compliance was available in 124/173 (71.7%) respondents. Ninety-six/124 (77.4%) reported to check the patient's compliance during therapy by verbal interrogation at each patient's visit. Visual inspection of urine color in patients on RMP was stated to be routinely done by 10/54 (18.5%). Directly observed therapy was reported to be arranged by 8/124 (6.4%).

Routine blood sampling was reported to be done in 166/173 (96.0%) before and in 163/173 (94.2%) during treatment. Frequency of requested specific analysis prior and during treatment were as follows: liver function tests 111/116 (95.7%) and 104/163 (63.8%), full blood count 103/166 (62.0%) and 92/163 (56.4%), creatinine 68/166 (41.0%) and 47/163 (28.9%), electrolytes 49/166 (29.5%) and 30/163 (18.4%) and C-reactive protein 39/166 (23.5%) and 9/163 (5.5%), respectively.

**Table 3. Reported dose and duration of antituberculosis agents in LTBI treatment.**

a)

| INH dose | | RMP dose | |
|---|---|---|---|
| ~5mg/kg/d | 9 (9.6%) | ~10mg/kg/d | 12 (22.2%) |
| ~10mg/kg/d | 50 (53.2%) | ~15mg/kg/d | 18 (33.3%) |
| ~15mg/kg/d | 1 (1.1%) | >15mg/kg/d | 5 (9.3%) |
| Other dose* | 15 (16.0%) | Other dose | 0 |
| Missing data | 19 (20.2%) | Missing data | 19 (35.2%) |
| total | 94 (100%) | total | 54 (100%) |

b)

| Treatment regimen / duration | INH monotherapy | INH / RMP combination therapy |
|---|---|---|
| 3 months | 7 (7.4%) | 19 (39.6%) |
| 4 months | 1 (1.1%) | 9 (18.8%) |
| 6 months | 6 (6.4%) | 2 (4.2%) |
| 9 months | 58 (61.7%) | 1 (2.1%) |
| Other time-span | 2 (2.1%) | 2 (4.2%) |
| No information | 20 (21.3%) | 15 (31.2%) |
| total | 94 (100%) | 48 (100%) |

Only response included that that stated to use INH and/or RMP for LTBI treatment.

Only those cases were included, that stated to use INH monotherapy or INH / RMP combination therapy for LTBI treatment.

Information on follow-up visits during LTBI treatment was available from 118/173 (68.2%) respondents: of these 4/118 (3.4%) reported no routine follow-up to be arranged; 14/118 (11.9%) only at the end of treatment, 16/118 (13.6%) reported arranging several follow-up visits. Information on chest x-rays performed at the end of treatment was available from 117 respondents: 63/117 (53.8%) routinely perform a chest-x-ray at the end of LTBI treatment, with 9/12 (75%) from Austria, 49/77 (63.4%) from Germany, and 5/28 (18%) from Switzerland, respectively. One hundred and twenty respondents provided information on routine visits following completion of LTBI treatment: in 32/120 (26.7%) no follow-up visit is reported, in 12/120 (10.0%) only in certain circumstances and 76/120 (63.3%) reported to arrange routine follow-up visits from 2–24 months after completion of LTBI treatment. This differed between Austria, Germany and Switzerland as follows: no follow-up visit in 1/12 (8.3%), 12/79 (15.2%) and 19/29 (65.5%), respectively, only in certain circumstances in 0/12, 7/79 (8.8%) and 5/29 (17.2%), respectively and routine follow-up visits in 11/12 (91.2%), 60/79 (75.9%) and 5/29 (17.2%), respectively.

## Exposure to multi-drug-resistant TB

Information on the management in case of exposition to an infectious case with multidrug-resistant (MDR) -TB (drug-sensitivity testing of the index case available) but no signs of infection (IGRA/TST negative) was provided by 115/173 (66.5%) of respondents. In 13/173 (7.5%) is the management was reported as "unknown" and in 45/173 (26.0%) no information was provided. In the remaining 115 respondents, 10/115 (8.7%) reported to always provide prophylactic treatment to the child, 17/115 (14.8%) only if the child was ≤ 5 years of age, 13/115 (11.3%) only if risk of infection was high regardless of the age of the child, 35/115 (30.4%) only in children ≤ 5 years of age and a high risk for infection, 13/115 (11.3%) reported they would use a "watch and wait" approach, and 27/115 (23.5%) would transfer the child to a TB

specialist. In case of suspected MDR-LTBI, 47/173 (27.2%) respondents did not provide an answer and in 13/173 (7.5%) the management was reported as "unknown". Forty-one/113 (36.3%) reported they would always start LTBI treatment, 12/113 (10.6%) only if the child was < 5 years of age, 3/113 (2.7%) only if the child was ≤ 5 years of age and had an increased risk for infection, 25/113 (22.1%) depending on risk and the patient's/ parent's wish, 30/113 (26.5%) would refer to a TB specialist and 2/113 (1.8%) would not give LTBI treatment at all.

In S2 Appendix information on availability of training on childhood TB is provided.

## Discussion

The survey highlights different aspects of paediatric LTBI diagnostics, treatment regimens and follow-up in three high-income, low-TB-incidence countries in central Europe.

The majority of the study respondents evaluate less than 50 children for TB per year bearing the risk of waning expertise for the management of childhood TB in low-incidence countries. Migration from high-TB-incidence countries to Europe continues with migrants being considered at risk for TB [1, 3]. Accordingly, around a third of respondents reported that more than half of children evaluated for TB were migrants that had recently arrived highlighting the relevance of TB in this group [23–25]

Both immunodiagnostic tests, TST or IGRA, are regarded as equal tools in the diagnosis of TB infection [26]. While sensitivity of both tests is comparable, IGRA tests are more specific with less cross-reactivity to non-tuberculous mycobacteria and *Bacillus Calmette Guérin* (BCG) vaccination [27]. In general, Quantiferon Gold tube test was reported to be used more frequently than the T-Spot TB-test, but differences between the countries were noted as practitioners from Switzerland chose T-Spot.TB about twice as often as colleagues from Germany and Austria. Although both IGRA tests are considered interchangeable, some evidence exists that T-Spot.TB might produce a lower rate of indeterminate results in children originating from the African continent and in immunocompromised children [28]. Information on immune status or origin of the children evaluated for TB was not assessed in our survey. However, studies have shown that IGRA tests are less sensitive in younger children with a higher rate of false negative or indeterminate results especially in infants compared to TST [29, 30]. Hence, the TST has been recommended as the standard of care immunodiagnostic test in children < 5 years of age by many experts [7, 31, 32]. Recently, there is growing evidence that new generation IGRA-tests are useful screening tools for TB infection in very young children some experts recommend including IGRAs in the diagnostic work-up even in the youngest age group [21, 33]. The importance of more solid evidence for the use of IGRA-tests in all age groups became particularly evident during the shortage of PPD RT 23 SSI availability in 2016 [22]. More than half of our respondents reported that they were affected by the BCG-shortage, and were forced to use an alternative tuberculin or an IGRA testing instead.

For the exclusion of active TB national and international guidelines recommend performing a chest x-ray in every child and adolescent with a positive TST or IGRA who had been exposed to an infectious TB index case [21, 26, 34]. In view of the reduced sensitivity of immunological tests combined with an increased risk for disease progression in children younger than 5 years of age, chest x-ray is often included in the initial diagnostic work-up in this age group [4]. Accordingly, almost 60% of the respondents reported to routinely perform a chest-x-ray together with an immunological test in younger children, while this approach was practiced in only 40% of children older than 5 years.

INH monotherapy was the first LTBI treatment regimen with proven efficacy against progression from LTBI to TB disease [35]. INH monotherapy is recommended to be given for 9 months -a duration that has shown optimal efficacy with limited hepatotoxicity in adults [36].

Shorter regimens using rifamycin-based regimens are increasingly preferred. This includes 3–4 months or RMP monotherapy, 3 months of INH and RMP combination therapy or once-weekly INH and rifapentine for 3 months, the latter not being available in Germany, Austria and Switzerland [37–39].

In our survey, INH monotherapy was the most frequent LTBI treatment reported to pre-scribed for a duration of 9 months with shorter duration reported in about 15%. Almost twice as many respondents preferred a longer INH monotherapy over a shorter INH-RMP combination therapy. A possible explanation might be concerns about adverse events. Hepatotoxicity is a well-known adverse effect of INH and RMP. The administered dose, TB disease severity, and being an "INH slow-acetylator" have been identified as risk factors for INH- or RIF-associated hepatotoxicity [40, 41]. However, a very low incidence of adverse events and an increased compliance with the shorter combination therapy compared to 9-months INH has been clearly demonstrated in children [42, 43].

Currently recommended daily doses for INH and RMP are INH 10 mg/kg and RMP 15 mg/kg bodyweight, respectively [11] and were increased from previous guidelines from INH 5mg/kg and RMP 10 mg/kg following pharmacokinetic studies in children [12–16]. In the current understanding, LTBI and TB disease are regarded as a continuum of the same disease ranging from asymptomatic infection to clinically active diseases [44, 45]. Therefore, there is no reason to assume that doses for LTBI and TB disease should be different. In our setting, reported dosages were below the recommended range in about a fifth of all respondents.

In order to detect adverse events timely, especially hepatotoxicity, monitoring of liver function during LTBI treatment is advisable. Regular follow-up visits have been recommended for low-incidence countries, while often not being feasible in high-incidence countries [21]. With well-adherent treatment in the context of an infection with a drug-sensitive organism, the risk of progression to disease is low [37, 46].

Nevertheless, in children the index-case or its susceptibility testing might not be available to the clinicians caring for the child or adherence to therapy was less than anticipated bearing an increased risk for treatment failure. Keeping this in mind, regular clinical follow-ups during treatment may increase adherence and detect early LTBI treatment failures. As there is no consensus on the routine use of a chest x-ray at the end of treatment and further follow-up visits with or without additional chest-x-rays after treatment completion, clinical practice is heterogeneous. In addition, follow-up visits especially in groups like refugees might not be feasible for reasons like relocation or return to the home country [2, 47].

In case of exposure to an MDR-strain or MDR-LTBI, the survey confirms further heterogeneity in the management. Numbers of children and adolescents with MDR-TB in Austria, Germany Switzerland are very low. International guidelines differ substantially for children being exposed or infected with MDR-TB with watchful waiting being recommended as well as different treatment regimen with second-line agents with assumed or known susceptibility [48–50]. Studies from South Africa and Micronesia provided evidence for the effectiveness of treatment (MDR-LTBI) and prophylaxis in MDR-TB contacts, with significantly less children progressing to active disease receiving treatment compared to those under clinical supervision only [49, 51]

Our study has limitations: the use of a questionnaire consisting mainly of questions with predefined answers limits the possibility to prescribe individual procedures. This survey was conducted immediately before national guidelines for paediatric TB were released and management of LTBI might have been improved in the meantime [21].

Nevertheless, our data highlights the heterogeneity in diagnostic, treatment and follow-up in paediatric LTBI patients in Austria, Germany and Switzerland. Dosing prescriptions of anti-TB drugs below the current recommended range of recommendations already issued in 2010 [52] in a substantial proportion of respondents is of concern.

Thus, regular and easily accessible educational activities and national up-to-date guidelines are key to ensure awareness and quality of care for children and adolescents with LTBI in low-incidence countries.

## Supporting information

**S1 Appendix. English version of the survey: Management of childhood and adolescent latent tuberculous infection (LTBI) in Germany, Austria and Switzerland.**
(PDF)

**S2 Appendix. Availability of training on childhood TB.**
(PDF)

## Acknowledgments

The authors would like to thank all participants for their valuable contributions to the study. In addition, all authors thank the German Society for Pediatric Infectious Diseases (DGPI) for supporting this study by facilitating access to the SurveyMonkey tool and Daniel Pieroth for his technical support. All authors are extremely thankful to Dr Bodo Königstein for his critical input and support of this study. Furthermore, all authors would like to pay homage to his unwavering and tireless life-long efforts in the field of tuberculosis in both children and adults in Germany. He is dearly missed.

## Author Contributions

**Conceptualization:** Ulrich von Both, Nicole Ritz, Matthias Bogyi, Folke Brinkmann, Stephanie Thee.

**Data curation:** Stephanie Thee.

**Formal analysis:** Philipp Gerlach, Stephanie Thee.

**Methodology:** Ulrich von Both.

**Supervision:** Ulrich von Both, Stephanie Thee.

**Writing – original draft:** Philipp Gerlach, Stephanie Thee.

**Writing – review & editing:** Ulrich von Both, Nicole Ritz, Matthias Bogyi, Folke Brinkmann.

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
