## [Decision Letter · Decision Letter 0]

6 Apr 2021

Management of childhood and adolescent latent tuberculous infection (LTBI) in Germany, Austria and Switzerland

PONE-D-21-00782

Dear Dr. Thee,

We’re pleased to inform you that your manuscript has been judged scientifically suitable for publication and will be formally accepted for publication once it meets all outstanding technical requirements.

Kind regards,

Joan A Caylà, PhD, MD

Academic Editor

PLOS ONE

1. We note that Figure 1 in your submission contain map images which may be copyrighted.

We require you to either (a) present written permission from the copyright holder to publish this figure specifically under the CC BY 4.0 license, or (b) remove the figure from your submission:

You may seek permission from the original copyright holder of Figure 1 to publish the content specifically under the CC BY 4.0 license. 

If you are unable to obtain permission from the original copyright holder to publish this figure under the CC BY 4.0 license or if the copyright holder’s requirements are incompatible with the CC BY 4.0 license, please either i) remove the figure or ii) supply a replacement figure that complies with the CC BY 4.0 license. Please check copyright information on all replacement figures and update the figure caption with source information. If applicable, please specify in the figure caption text when a figure is similar but not identical to the original image and is therefore for illustrative purposes only.

Reviewers' comments:

Reviewer's Responses to Questions

**Comments to the Author**

1. Is the manuscript technically sound, and do the data support the conclusions?

Reviewer #1: Yes

2. Has the statistical analysis been performed appropriately and rigorously? 

Reviewer #1: Yes

3. Have the authors made all data underlying the findings in their manuscript fully available?

Reviewer #1: Yes

4. Is the manuscript presented in an intelligible fashion and written in standard English?

Reviewer #1: Yes

5. Review Comments to the Author

Reviewer #1: As the global focus on the importance of addressing latent tuberculosis is scaled up as part of the Stop TB strategy, this survey of practices pertaining to the diagnosis and treatment of latent TB among children and adolescents in a low incidence country is a clear assessment of the needs that to be instituted in order to advance our goal of TB elimination - continuing educational activities regarding updated guidelines and best practices for pediatric latent TB infection.

6. PLOS authors have the option to publish the peer review history of their article (what does this mean?). If published, this will include your full peer review and any attached files.

Reviewer #1: **Yes: **Alfred A. Lardizabal

---

## [Editor Report · Acceptance letter]

19 Apr 2021

PONE-D-21-00782 

Management of childhood and adolescent latent tuberculous infection (LTBI) in Germany, Austria and Switzerland 

Dear Dr. Thee:

I'm pleased to inform you that your manuscript has been deemed suitable for publication in PLOS ONE. Congratulations! Your manuscript is now with our production department. 

Kind regards, 

on behalf of

Professor Joan A Caylà 

Academic Editor

PLOS ONE